# Oncolytic Virotherapy: A New Paradigm in Cancer Immunotherapy

**DOI:** 10.3390/ijms25021180

**Published:** 2024-01-18

**Authors:** Simona Ruxandra Volovat, Dragos Viorel Scripcariu, Ingrid Andrada Vasilache, Cati Raluca Stolniceanu, Constantin Volovat, Iolanda Georgiana Augustin, Cristian Constantin Volovat, Madalina-Raluca Ostafe, Slevoacă-Grigore Andreea-Voichița, Toni Bejusca-Vieriu, Cristian Virgil Lungulescu, Daniel Sur, Diana Boboc

**Affiliations:** 1Department of Medical Oncology-Radiotherapy, “Grigore T. Popa” University of Medicine and Pharmacy, 16 University Str., 700115 Iasi, Romania; simonavolovat@gmail.com (S.R.V.); madalina.ostafe@gmail.com (M.-R.O.); slevoacavoichita@yahoo.com (S.-G.A.-V.); tonibejusca@gmail.com (T.B.-V.);; 2Department of Surgery, “Grigore T. Popa” University of Medicine and Pharmacy, 16 University Str., 700115 Iasi, Romania; dscripcariu@gmail.com; 3Department of Obstetrics and Gynecology, “Grigore T. Popa” University of Medicine and Pharmacy, 700115 Iasi, Romania; 4Department of Biophysics and Medical Physics—Nuclear Medicine, “Grigore T. Popa” University of Medicine and Pharmacy, 16 University Str., 700115 Iasi, Romania; catistolniceanu@yahoo.com; 5Department of Medical Oncology, Al. Trestioreanu Institute of Oncology, 022328 Bucharest, Romania; iolanda.augustin@gmail.com; 6Department of Radiology, Grigore T. Popa University of Medicine and Pharmacy, 700115 Iasi, Romania; cristian.volovat@yahoo.com; 7Department of Oncology, University of Medicine and Pharmacy of Craiova, 200349 Craiova, Romania; cristilungulescu@yahoo.com; 811th Department of Medical Oncology, “Iuliu Hatieganu” University of Medicine and Pharmacy, 400347 Cluj-Napoca, Romania; daniel.sur@umfcluj.ro

**Keywords:** virus engineering, immune virotherapy, viral biotechnology, oncolytic virotherapy, nanomedicine

## Abstract

Oncolytic viruses (OVs) are emerging as potential treatment options for cancer. Natural and genetically engineered viruses exhibit various antitumor mechanisms. OVs act by direct cytolysis, the potentiation of the immune system through antigen release, and the activation of inflammatory responses or indirectly by interference with different types of elements in the tumor microenvironment, modification of energy metabolism in tumor cells, and antiangiogenic action. The action of OVs is pleiotropic, and they show varied interactions with the host and tumor cells. An important impediment in oncolytic virotherapy is the journey of the virus into the tumor cells and the possibility of its binding to different biological and nonbiological vectors. OVs have been demonstrated to eliminate cancer cells that are resistant to standard treatments in many clinical trials for various cancers (melanoma, lung, and hepatic); however, there are several elements of resistance to the action of viruses per se. Therefore, it is necessary to evaluate the combination of OVs with other standard treatment modalities, such as chemotherapy, immunotherapy, targeted therapies, and cellular therapies, to increase the response rate. This review provides a comprehensive update on OVs, their use in oncolytic virotherapy, and the future prospects of this therapy alongside the standard therapies currently used in cancer treatment.

## 1. Introduction

The general perception of viruses as malefic devils, owing to the often fatal infections they cause in the human body, started changing in recent decades with the realization of the possibility of turning them into protective angels that can be used as weapons against malignant cells. The idea of using viruses for cancer treatment arose with observations of the association between tumor regression and viral infections documented in some case reports [1].

In recent years, oncolytic viruses (OVs) have attracted particular interests in view of their ability to selectively attack and destroy tumor cells and potentially stimulate antitumor immunity. There are many advantages in engineering tumors that selectively target OVs with a manageable safety profile and efficacy against different cancers. A candidate virus selected to destroy tumor cells must possess some hallmarks: it must be proimmunogenic, should not cause chronic infectious disease, is safe for various human populations, should exert lytic activity in tumor cells, and should have the capability to integrate into the host genome [2]. Proimmunogenicity of an oncolytic virus is characterized by the following: the existence of antigenicity to be recognized by the immune system, infectivity, replication to trigger a robust immune response, and genetic variation that can contribute to immune recognition and inflammatory response.

Genetic engineering plays an essential role in developing OVs with a manageable safety profile and high efficacy against a wide range of cancers. In this regard, OVs must be tailored specifically for the respective tumor type and tumor mutations by modifying the noncoding mutations and adding or removing certain functions, genes, and noncoding elements to confer additional beneficial properties on them [3]. There are three research directions for cancer treatment using OVs: viral replication, tumor growth, and immune activation [3]. Consequently, the main targets of OVs are tumor cells, host immunity, and the tumor microenvironment. Most viral vectors initiate immunogenic cell death (ICD) by releasing tumor-associated antigens, which are followed by antitumor immune responses. By activating antitumor immunity, OVs exceed the performance of ICIs and other precisely targeted drugs. OVs exhibit a wide range of antitumor immune-boosting properties. Patients with suitable molecular profiles must be identified to benefit from OV therapy. Therefore, the discovery of predictive biomarkers for OV treatment is important.

Here, we review various aspects of OVs and issues related to their use in oncolytic virotherapy. We believe that oncolytic virotherapy represents a new paradigm in the immunotherapy of cancer. 

## 2. Attractive Viruses for Oncolytic Virotherapy

Virotherapy, a therapeutic method based on the introduction of OV-infected cells into a host diagnosed with a malignancy, could be an innovative futuristic approach to cancer therapy. Recent studies have revealed that the use of OVs leads to specific destruction of tumor cells by inducing systemic antitumor immunity [3]. However, the process is not as simple as it appears because not all cell types and viruses can be used for this purpose. Understanding the tumor tropism of different types of viruses for selected tumors, as well as the combined therapies in which they can be included, determines the selection of the appropriate and accurate option for viral oncotherapy aimed at increasing the overall effectiveness of the treatment. OVs can be single-stranded or double-stranded RNA or DNA viruses and are classified into natural viral strains and genetically modified viruses.

The dominant types of OVs eligible as candidates for viral oncotherapy are described below with main characteristics (Table 1) and with benefits and issues (Table 2):

The herpes virus (HSV), comprising two serotypes (HSV-1 and HSV-2), is an enveloped virus with a nucleocapsid that protects a double-stranded DNA [4,5,6]. The advantages of the HSV as an oncolytic virus are as follows [7,8,9,10,11,12,13]: (i) a large genome and complex structure that enables the insertion of large fragments and multiple transgenes without limiting the viral efficiency; (ii) an ability to infect most types of malignant cells; (iii) flexibility in inserting multiple transgenes; (iv) despite the high frequency of anti-virus antibodies in a large proportion of the population, they do not prevent virus replication. The binding of the HSV to various cellular receptors is ensured by four viral glycoproteins, namely gB, gD, gH, and gL [14].

Talimogene laherparepvec (T-VEC), based on a genetically modified HSV, which encodes the human granulocyte-macrophage colony-stimulating factor (GM-CSF), is the only one among the seven oncolytic HSVs approved by the FDA and the EMA for the treatment of IIIB/C–IVM1a melanoma. T-VEC stimulates both local and systemic antitumor immune responses. T-VEC has been evaluated in two phase II clinical studies—in locally advanced cutaneous angiosarcoma in one and soft tissue sarcoma in the other—in combination with radiotherapy (NCT02923778) [11]. OH2 is another genetically engineered type 2 oncolytic HSV that expresses the GM-CSF to promote antitumor immune responses [15,16]. G207 is an engineered herpes virus with a gene deletion in regions ICP 34.5 and ICP 6 with a direct oncolytic effect and can also stimulate a systemic antitumor immune response. An engineered virus with three genome modifications derived from G207 is G47Δ, which is highly replicative in tumors and has more cytopathic effects [17].

The adenovirus (Ad), with a double-stranded DNA genome and an icosahedral capsid, has a C-terminal responsible for identifying cellular receptors [18]. The advantages of adenovirus oncolytic therapies are the following: the virus also destroys the cancer stem cells, triggering the various kinds of cell death signaling and releasing many damage-associated molecular pattern molecules (DAPSs) such as high-mobility box 1 (HMGB1), heat-shock proteins (HSPs), and nucleic acids. Adenoviruses induce local inflammation responses, a high production of viral particles, and a release of tumor antigens, having direct and indirect oncolysis effects [19].

Most oncolytic adenoviruses (oAds) infect cells in combination with the coxsackievirus and adenovirus receptor (CAR). Ads are suitable candidates for oncolytic viral therapy and are among the most studied viruses in this field owing to their high genetic stability and availability. Genetic stability refers to the ability of the virus to maintain its genetic material, specifically the therapeutic or desired genes, without undergoing significant mutations or alterations over time. In 2005, the Chinese FDA approved the use of Oncorine, the first oAd, together with chemotherapy, for the treatment of nasopharyngeal carcinoma [20]. As of 2021, the Human Adenovirus Working Group has identified 104 registered adenovirus genotypes, 18 of which are currently being tested for viral oncotherapy, with 59 trials conducted from 2005 to 2021. For oncolytic therapy to be effective, viruses need to be genetically engineered, and nonessential genes must be replaced with other genes that enhance the therapeutic effect [19].

There are three types of genetically engineered adenoviruses, also known as conditionally replicating adenoviruses (CRAd), which are under investigation [19]. The first generation of CRAd is represented by Onyx-015, which binds and inactivates p53 and Ixovex-1 that prevents tumor growth, and is involved in different clinical trials [21]. The second generation of CRAd contains an engineered adenovirus Delta-24-RGD (DNX-2401) that targets the tumor suppressor pathway of retinoblastoma (Rb). The third generation of CRAd is represented by a virus similar to DNX-2401 with changes in the E1A promoter to increase the performance in tumor suppression [22].

Maraba, a rhabdovirus, was recently considered a suitable candidate for oncolytic viral therapy [16]. The idea of using this virus stemmed from the use of a similar virus—the vesicular stomatitis virus—in oncolytic virotherapy trials, and this virus has a more potent oncolytic activity. In vitro, with normal cells, the Maraba virus shows a very broad oncotropism, good tumor selectivity, and low virulence. In addition, in vivo studies performed in xenograft models have shown strong antitumor activity and superior therapeutic efficacy (Table 3) [23].

The morbillivirus (measles, MeV) is an RNA paramyxovirus with tropism for a broad range of cancer types, including ovarian and breast cancers, multiple myeloma, mesothelioma, and glioblastoma [19]. MeV oncolytic therapy is being tested in combination with other therapies, such as chemotherapy for peritoneal or ovarian cancer [19,24].The interactions of MeVs with host cells are mediated by three receptors, namely, Nectin-4, CD46, and SLAM/CD150. Systemic administration of MeV could be difficult owing to the presence of preexisting antimeasles antibodies, which is why several strategies have been developed, such as the use of a cellular vehicle to transport the virus, coding of genes to suppress the immune system, replacement of some glycoproteins with nonreactive ones, or administration in combination with immunosuppressive drugs [19].The MeV has a favorable safety profile and limited toxicity, and MeV-specific T-cell responses may increase the effectiveness of oncolytic virotherapy (Table 3) [25].

The Newcastle disease virus (NDV) is an avian paramyxovirus type 1 virus that is naturally found in birds. It has been used in oncolytic therapy research for five decades; it is highly safe because it does not go through the process of gene exchange through recombination [26].

This is evident from the fact that humans are nonpermissive hosts of this virus. The NDV, through its RIG-I receptors, can slow down tumor activity and has a very broad oncotropism, including that for breast carcinoma, colorectal cancer, and head and neck carcinoma [27].

The reovirus is frequently found in the environment, and its anticancer activity can be potentiated by eradicating virally infected tumor cells. Although the reovirus has been studied as a therapeutic option for melanomas, myelomas, and gliomas, trials (Table 3) are also being conducted for other malignancies, such as pancreatic cancer, lung cancer, and colorectal cancer (GOBLET study in combination with pelareorep and atezolizumab) [28].

The coxsackie virus has oncotropism for bladder cancer, in particular. Some researchers have investigated the oncolytic activity of coxsackievirus B3 for the treatment of colorectal carcinomas and showed significant inhibition of tumor growth but with some serious side effects, such as myocarditis and pancreatitis [29]. In a phase II clinical trial in patients with unresecable melanoma, with coxsackievirus A21 (V937) injected intratumorally, the 6-month progression-free survival was 38.6% (95% CI, 26.0 to 52.4) (Table 3) [30].

The parvovirus is the smallest virus clinically investigated for cancer virotherapy. Three important approaches have been investigated for improving the oncolytic activity of this virus, viz., combination with various other anticancer agents, preparing novel parvovirus with upgraded oncolytic and immunomodulatory functions, and using cellular compounds as biomarkers to monitor the response to parvovirus treatment [31] (Table 3).

**Table 1 ijms-25-01180-t001:** Selected oncolytic viruses—Main characteristics.

Characteristics	DNA			RNA				
	Adenovirus	Herpes Simplex Virus-1	Vaccinia Virus	Measles	Vesicular Stomatitis Virus	Newcastle Disease Virus	Maraba Virus	Reovirus
Genome size (nm)	dsDNA 70–100	dsDNA 200	dsDNA 70–100	ss(-)RNA 100–200	ss(-)RNA 80	ss(-)RNA 100–500	ss(-)RNA 80–95	dsRNA 60–80
Capsid symmetry	Icosahedral	Icosahedral	Complex	Icosahedral	Helical	Helical	Helical	Icosahedral
Envelope	Naked	Enveloped	Complex coat	Enveloped	Enveloped	Enveloped	Enveloped	Naked
Site of replication	Cytoplasm and nucleus	Cytoplasm and nucleus	Cytoplasm	Cytoplasm	Cytoplasm	Cytoplasm	Cytoplasm	Cytoplasm
Entry receptor	CD46, CAR	Nectin 1,2, HVEM	Without specific receptor	CD46, SLAM	LDRL	Sialic acid	LDRL	Without specific receptor
Ref.	[7,32,33]	[7,8,32,33]	[8,9,10,11]	[8,9,10,11]	[8,9,10,11]	[12]	[8,9,10,11]	[13]

**Table 2 ijms-25-01180-t002:** Selected oncolytic viruses—Benefits and issues.

Characteristics	DNA			RNA				
	Adenovirus	Herpes Simplex Virus-1	Vaccinia Virus	Measles	Vesicular Stomatitis Virus	Newcastle Disease Virus	Maraba Virus	Reovirus
Treated cancer type	Brain	Skin	Liver	Breast, ovary	Solid tumors	Solid tumors	NSCLC	Myeloma
Pathogenicity of native virus	Fever, respiratory acute distress, gastroenteritis	Gingivo-stomatitis, kerato-conjunctivitis, encephalitis	Severe pneumonia	High fever, red rash, pneumonia, encephalitis	Oral vesicles/ulcers Skin rashes, fever	Hemorrhagy of digestive tract, conjunctivitis	Flu-like illness	Respiratory tract infection, gastroenteritis, diarrhea
Benefits	Produce high viral concentrations, accessible, genetic manipulation, strong lytic activity, potentiate immunomodulatory agents in combination therapy	Large genome, suitable for genetic modification replication only in cells	Spread fast and efficiently, rapid life cycle pace, high insertion capacity, risk of latent infection, side effects	Promising results of clinical trials	Do not infect humans, rapid life cycle pace	Low immunogenicity in humans, multicentric replication, spread fast	-	Intravenous administration, high dosed without high toxicity
Issues	Tropism for a wide variety of tissues, attenuated viral spread	Possibility of latent infection, high pathogenicity	Pathogenicity	Pathogenicity	Issues with gene editing	Issues with gene editing, systemic toxicity		Issues with gene editing
Type of ICD induced	Autophagy, necroptosis	Autophagy	Necroptosis	Apoptosis	Apoptosis	Apoptosis, autophagy, necroptosis	Apoptosis	Apoptosis, necroptosis
Blood–brain barrier penetration	−	−	+	−	+/−	−	−	+
Ref.	[7,32,33]	[7,32,33]	[8,9,10,11]	[8,9,10,11]	[8,9,10,11]	[12]	[8,9,10,11]	[13]

**Table 3 ijms-25-01180-t003:** Selected current clinical trials with oncolytic viruses (www.ClinialTrials.gov accessed on 2 January 2024).

Class of Oncolytic Virus	Oncolytic Virus	Phase Trial	Type of Cancer	Primart Endpoint	Current Status	ClinicalTrials.gov ID
Herpes simplex virus	OH2	II	Bladder advanced	Objective Response Rate	recruiting	NCT05248789
	OH2	Ib/II	Pancreas second line	objective response rate	recruiting	NCT04637698
	OH2	II	Melanoma third-line	Overall survival	recruiting	NCT05868707
Adenovirus	DNX-2401	II	glioblastoma	Overall survival, time to tumor response	completed	NCT02798406
	OBP-301	II	Head and neck squamous, recurrent, or progressive	Overall response rate	completed	NCT04685499
rhabdovirus	MG1 maraba	II	Solid tumors, advanced	objective response rate	Active, not recruiting	NCT02285816
morbillivirus	Measles (MV-NIS)	I/II	Ovarian, recurrent	12 moths overall survival	recruiting	NCT02068794
reovirus	Reolysin	II	Melanoma metastatic	Tumor response	completed	NCT00651157
Picornavirus	Coxsackie virus A21	II	Melanoma Stage IIIC/IV	6 months progression-free survival	completed	NCT01227551
parvovirus	ParvOryx	II	Pancreas advanced	Safety and tolerability	completed	NCT02653313

## 3. Anticancer Mechanisms of Oncolytic Virotherapy 

### 3.1. Selection of an OV

Ideal OVs must meet the following requirements: (i) High targeting of tumor cells; (ii) Rapid replication and expansion; (iii) Low immunogenicity with attenuated antiviral responses; (iv) Strong activation of the immune response. 

Viral replication and expansion involve a series of processes by which viruses make copies of themselves and increase their numbers within a host organism. The specific mechanisms can vary among different types of viruses, but here is a general overview: 1. Attachment and Entry, when viruses first attach to specific host cells using surface proteins and then enter the host cell, either by direct fusion with the cell membrane or by endocytosis. 2. Release of Viral Genome. Once inside the host cell, the virus releases its genetic material (either DNA or RNA) into the cell. 3. Replication and Transcription. The viral genome takes control of the host cell’s machinery to replicate and transcribe its genetic material. DNA viruses often replicate in the host cell nucleus, while RNA viruses replicate in the cytoplasm. 4. Translation and Protein Synthesis: The host cell’s ribosomes are hijacked to synthesize viral proteins using the instructions from the viral RNA or DNA. 5. Assembly. Newly synthesized viral components (proteins and genetic material) are assembled to form complete viral particles. 6. Maturation, where the assembled viral particles undergo maturation and where they acquire the ability to infect other cells. 7. Release. The mature viruses are released from the host cell, either by cell lysis (cell bursting) or by budding from the cell membrane. 8. Spread and Expansion. Released viruses can infect neighboring cells, repeating the replication cycle and leading to the expansion of the viral population within the host [10,32].

Low immunogenicity with attenuated antiviral responses refers to a situation where the immune system exhibits a reduced or weak response to a virus, particularly when the virus has been attenuated or weakened for various purposes, such as vaccine development. Attenuated antiviral responses involve using a weakened form of a virus as a vaccine. The virus is modified to reduce its virulence, making it less harmful, while still retaining the ability to stimulate an immune response. This approach aims to induce an immune response that provides protection against the natural, more potent form of the virus. An example of an attenuated oncolytic virus is T-VEC. 

When identifying an OV that destroys malignant cells, the first consideration is the viral skeleton. DNA viruses have multiple advantages in that they have a more stable genome and lend themselves to genetic engineering more easily than RNA viruses [34,35]. DNA viruses have a lower immunogenicity than RNA viruses and have more repair mechanisms, reducing the likelihood of errors during replication. Some DNA viruses, such as retroviruses, have the ability to integrate their genetic material into the host genome.

RNA viruses are smaller in size, cross the blood–brain barrier, and have more constant replication than DNA viruses; however, they have higher genetic instability and mutation rates than DNA viruses. The most commonly used platforms are those with DNA viruses, such as HSV type-1 and Ads [25,36,37]. Genetic engineering is an alternative to enhance certain properties of OVs. Two-thirds of viruses in clinical trials have been reported to be genetically engineered [37]. The first attempts to genetically engineer OVs were made in 1991 using HSVs. Today, the fourth generation of genetically engineered oncoviruses is available for use in tumor immunotherapy [38]. Research on genetically engineered OVs has focused on increasing their efficacy and safety, mainly to promote selective replication, decrease immunogenicity, promote tumor destruction, and attenuate viral pathogenicity. More than 20 virus families have been genetically engineered to date; these include MeVs, NDVs, reoviruses, polioviruses, vaccinia viruses (VVs), HSVs, Ads, poxviruses, myxomaviruses, and vesicular stomatitis viruses (VSVs). The purpose of these transformations is to target viral replication, initiation of immune stimulation, and release of tumor-associated antigens [39,40,41,42,43]. Some preclinical studies have demonstrated that engineered OVs play an immunomodulatory role by inducing the desired immune response. An example is the stimulation of the host immune response by the addition of the GM-CSF or ICAM-1 transgenes [35]. A representative example is the VSV, in which glycoproteins in the envelope have been modified to decrease the neurotoxicity of the virus [44]. Various strategies have been developed for the genetic engineering of OVs. It is mandatory to understand the biology and genetics of viruses to understand the relationships between the virus and the host and to comprehend how cells defend themselves against viral infections. Genetic engineering uses transgenes with different mechanisms to destroy neoplastic cells. Different strategies are employed to substitute different proteins in viruses to increase tumor targeting [45]. Many viral receptors are highly expressed by cancer cells that permit higher viral uptake in cancer cells than in normal ones. Some receptors, such as laminin, the CAR, CD 46, and CD155, are overexpressed in different cancer cells, resulting in increased uptake of Ads, Sindbis viruses, and Poliviruses [46,47,48,49,50]. Some viral proteins are toxic for neoplastic cells and can directly kill cells at the time of virus exposure, before viral replication, as E3 death protein and E4 or f4 proteins encoded by Ads [51]. In one study, glycoprotein G from the VSV was substituted with a variant protein from a choriomeningitic lymphocytic virus to increase the selectivity of the VSV for malignant cells [52]. Another strategy is to increase the host response to tumors. The T-VEC vaccine contains an Ad genetically modified to express the GM-CSF and increase the recruitment of antigen-presenting cells (APCs) into the tumor immunosuppressive microenvironment, resulting in the recruitment of various cytokines that amplify the antitumor activity. The efficacy of drug delivery is related to the delivery of the drug to the tumor bed without damaging normal cells and keeping the drug undegraded until it reaches the tumor. In one study, an oncolytic Ad coated with human albumin prolonged the survival rate of the virus [53]. In another study, the human albumin coating of an adenovirus protected it from neutralizing antibodies [54].

### 3.2. Antitumoral Mechanisms of OVs

Various mechanisms by which OVs act against tumor cells have been described (Figure 1). 

OVs, naturally, or due to genetically engineered modifications, exhibit tumor selectivity, replicate preferentially in cancer cells by targeting receptors on the surface of malignant cells, and cause intracellular aberrations within pathways and gene expression in cancer cells but do not affect healthy cells [59,60]. OVs can infect healthy cells, but they exhibit mechanisms for inhibiting viral spread, which are usually deficient in cancer cells [40]. Thus, in a VSV infection, normal cells can produce type I IFN to prevent viral replication, but most cancer cells have deficient type I IFN signaling [61,62]. It was shown that the Semliki virus infection induces, in tumor cells, the secretion of WNT ligands (WNT2B and WNT9B) and the stabilization of β-catenin (CTNNB1), and it decreases the expression of IFNB1 in a feedback mechanism, favoring the attenuation of host antiviral response [63]. Some studies have reported that two mouse cell lines (NR6 and B82), expressing no epidermal growth factor receptors (EGFR), were relatively resistant to reovirus infection, but the same cell lines transfected with the gene encoding EGFR express significantly higher susceptibility. This enhancement of infection efficiency requires a functional EGFR since it was not observed in cells expressing a mutated EGFR as in tumor cells [64,65]. Oncogenic viral infections, such as HPV, HCV, and KSHV, have developed mechanisms to attenuate tumor suppressor genes, such as p53 and RB, during infection to promote viral proliferation, emphasizing the importance of these gatekeeper genes in controlling viral infection [66,67,68].

The main antitumoral mechanisms employed by OVs are as follows: (i) Direct tumor lysis; (ii) TME remodeling with activation of antitumor activity; (iii) Gene targeting by OVs; (iv) Disruption and remodeling of the vascular system at the tumor site; (v) metabolic reprogramming of tumor cells; and (vi) immunogenic cell death (ICD) induced by OVs [8,9,58].

#### 3.2.1. Direct Tumor Lysis

A classic antitumor mechanism employed by OVs is replication within cancer cells, followed by direct lysis of these cells; this mechanism is termed direct virus-induced oncolysis (Figure 2) [40]. Consequently, transgenes are expressed and trigger host immune responses, thereby activating antitumor immunity [69].

It has been reported that an engineered Newcastle disease virus leads to the loss of mitochondrial membrane permeability in tumor cells, followed by ICD [70]. The M1 virus induces endoplasmic reticulum stress-mediated apoptosis, followed by cytolysis of tumor cells [71]. The engineered virus, A4, which carries the TRAIL gene and induces the expression of the TRAIL protein on the viral surface, can bind with TRAILR1 or TRAILR2 receptors on tumor cells and promote apoptosis [72]. This mechanism induces tumor cell death locally by direct tumor lysis, but it also triggers a systemic response by activating the immune system (Figure 2D) and activates the endothelial cells involved in the vascularization of the tumor, reducing tumor angiogenesis [73].

#### 3.2.2. TME Remodeling with the Activation of Antitumor Activity

A complex mechanism of the antitumor activity of OVs is TME remodeling, with the activation of antitumor activity, local OV-induced inflammation, and indirect mediation of apoptosis in both infected and uninfected tumor cells. 

The cytolysis of tumor cells releases tumor antigens and leads to a systemic antitumor-immune response in the host. TME is an active support network composed of immune cells, vasculature, and dysfunctional metabolic pathways that stimulate the proliferation of tumor cells. Some environments are considered immunologically “cold” due to reduced levels of tumor antigens and the infiltration of tumor suppressor immune cells. OVs are powerful agents of immunotherapy that can affect the immunosuppressive environment and create an immunologically “hot” environment that favors the action of the host immune system. OVs mediate the remodeling of the TME, creating an environment that favors the influx and activation of antitumor immune cells, favoring tumor elimination [74,75,76] (Figure 3).

OVs undergo replication and amplification, which are major factors in tumor destruction, leading to the infection of new tumor cells. These replication and amplification processes are specific only to OVs and are not found in other nonbiological cancer therapies [77].

DC/T-cell activation

The antitumor immune activity is triggered by OVs through various mechanisms, including innate and adaptive immunity and the induction of ICD. 

Releasing the DAMPs. OVs lead to endoplasmic reticulum (ER) stress, followed by apoptosis, the release of viral pathogen-associated molecular patterns (PAMPs), tumor-associated antigens, damage-associated molecular patterns (DAMPs), and the activation of innate immunity. PAMPs, together with viral elements, bind to Toll-like receptors (TLRs), inducing the maturation of dendritic cells (DCs) and the release of cytokines [78]. The ICD activated by OVs can be followed by the release of other DAMPs, such as calreticulin (CRT), ATP, heat shock proteins (HSPs), uric acid, high mobility group box 1 protein (HMGB1), and cytokines (IFN-α, IFN-γ, tumor necrosis factor-α, and interleukins 1, 6, 8, and 12) [79]. 

The activation of DCs. DAMPs can also initiate the maturation of DCs with the support of ligands for TLR4 on DCs (HMGB1 and CRT). DCs are rarely present in tumors but represent strong, professional APCs related to the adaptive and innate immune systems. The modulation of DC populations in the TME through oncolytic effects provides novel tumor antigens and supports the infiltration and maturation of DCs. Antigen presentation to CD4^+^ and CD8^+^ T cells is followed by the activation of effector T-cell responses, which memorize information, proliferate in the lymphatic nodes, and migrate into the tumor [73]. 

Genetic engineering has been used to modify oncolytic viruses by deleting or adding payloads (monoclonal antibodies, cytokines, and chemokines) to increase the immune response in the TME and limit the adverse effects. One such example is the addition of the GM-CSF, a cytokine that triggers the transformation of DCs into powerful APCs. Preclinical data demonstrated the existence of an abscopal effect of GM-CSF-armed HSVs [80,81]. IL12 is also involved in the maturation of DCs, T, and natural killer (NK) lymphocytes in the TME, the inhibition of regulatory T cells (Tregs) and MDSC activities, and the induction of M1 polarization of macrophages [82]. A major issue regarding IL12 is its association with lethal inflammatory syndrome due to systemic accumulation of IL12. To avoid these toxicities, an oAd expressing a mutant form of IL12 was engineered for safe administration [83].

Involvement of macrophages

The presence of M2 macrophages in the TME is a powerful prognostic factor for low overall survival in most cancers [84]. At the TME level, OVs act on M2 macrophages, which are repolarized to an M1 phenotype, and express various proinflammatory cytokines and chemokines (IFN-γ, CXCL10, various interleukins), among others, that reinforce the antitumor immune environment; this effect has been shown in some studies on OVs, including VSVs, HSVs, reoviruses (RV), paramyxoviruses (MeV and MuV), GLV-1h68 and Ad5 D24-RGD [74,85,86,87,88,89,90]

Involvement of NK cells

NK cells are important mediators of oncolytic virotherapy. Also, NK cells are involved in the fight against viral infections, and therefore, they can have a detrimental effect on the efficacy of OVs. The effects of OVs on NK cells are promoting NK cell recruitment from the circulation to the tumor, enhancing NK cell activation, and preventing and reversing NK cell anergy. OVs have the capacity to lyse tumor cells and trigger immunogenic cell death, followed by the releasing of DAMPs that are detected by DCs that promote inflammation. In addition, OVs infect DCs, and as a response to infection, tumors, and myeloid cells produce chemokines that recruit NK cells into the tumor. Also, DCs and other myeloid cells produce various cytokines that enhance NK cell survival and prevent and reverse NK cell anergy. When NK cells are activated, they produce a high amount of IFN-γ and kill tumor cells. NK cell ligands at the cancer cell surface are also modulated by OVs. In addition, OV-infected cells release viral proteins that are recognized by NK cells, which increase the expression of activating receptors and enhance the killing of tumor cells [91].

Some OVs mediate an increased production of cytokines and chemokines that promote the activation of NK cells, followed by the expression of NK cell-inhibitory ligands, such as MHC-I, in tumor cells [92,93].

It was demonstrated that the IL2-armed Sendai virus increased the recruitment of NK cells in a mouse model of angiosarcoma [94]. In a different study, an IL-2-expressing parvovirus (H-1PV) increased the activation of NK cells in a pancreatic ductal adenocarcinoma model [95].

An engineered *Vaccinia virus* (VV), VVΔTKΔN1L, armed with IL12 and with deletion of thymidine kinase and N1L genes, prolongs survival when used as a neoadjuvant treatment in various murine and hamster surgical models of cancer [96].

Involvement of neutrophils

The antitumoral effects of VV- and VSV-based viruses that promote apoptosis in tumor cells were demonstrated [68,97,98]. In preclinical experiments, it was found that TGF-β-polarized N2 neutrophils have a protumoral effect, whereas type-1 IFN-polarized N1 neutrophils have proimmune and antitumoral outcomes. N1 promotes the recruitment and activation of CD8^+^ T cells and the production of cytokines, such as VGF, TNF-α, the GM-CSF, and IL12. A recombinant poliovirus vaccine strain, named PVSIPO, which includes a heterologous internal ribosomal entry site, was shown to promote neutrophil accumulation in prostate and breast cancers and was correlated with improved survival [99].

Involvement of Tregs

Tregs also induce strong immunosuppressive activity by hyperexpressing the transcription factor, Foxp3. Infection with AdCMVΔ24 was demonstrated to decrease the Treg levels in the TME [100,101,102,103]. Recent studies reported that oncolytic VV could directly infect and decrease the number of Tregs in murine models of cancer, followed by antitumoral effects [101,102,103].

Stromal modifications

Some OVs target the tumor stroma and remodel the extracellular matrix (ECM) to improve their antitumoral effects. An oAd expressing relaxin combined with various modulators forming a platform oAd/IL-12/GM-RLX, which caused ECM degradation, enhanced the penetration of therapeutic monoclonal antibodies and promoted antitumor effects in pancreatic cancer models. Moreover, an oAd expressing PH20 hyaluronidase caused hyaluronan degradation in the ECM, which improved its spread [104,105,106,107]. Cancer-associated fibroblasts (CAFs) are upregulated in many cancers and activate fibroblasts from the stroma via fibroblast activation protein (FAP). Tumor-associated pericytes are the main stromal cell subset in glioblastoma models. An oAd, ICOVIR15, decreased the number of FAP^+^ pericytes, glioblastoma cells, and GMP-associated stromal FAP^+^ murine cells [108]. Another oncolytic VV, Western reserve double-deletion (WRDD), has been demonstrated to target CAFs and to have antitumor effects in melanoma [109].

#### 3.2.3. Gene Targeting by OVs

Gene targeting is a major antitumor mechanism of OVs. For example, reoviruses exploit specific cellular characteristics that are often altered in cancer cells.

In tumor cells, oncogenes are activated, and tumor suppressor genes are inactivated, which results in the modulation of the signaling of specific proteins, such as *TP53*, *PTEN*, *RB1*, and *RAS*, and activation of tumorigenesis, angiogenesis, inhibition of apoptosis, invasion, and metastasis. These signaling cascades are targeted by OVs, which modulate various signaling pathways in tumor cells. The action of reoviruses on tumor cell cultures has been studied, and viral selective replication steps inside the tumor cells were identified—*Ras* mutation can promote reovirus entry and oncogenic transformation, enhance the production of OVs, induce apoptosis, and prevent interferon production after reovirus infection, allowing the spread of the virus due to defective antiviral response [110]. Reovirus gains entry into cells by interacting with specific cell surface receptors. Changes in the expression or accessibility of these receptors on cancer cells may enhance the virus’s ability to enter and infect them. Oncolytic VVs and VSVs can only infect tumor cells and promote endothelial destruction due to the hyperexpression of ras/mitogen-activated protein kinase (*MAPK*) [111]. There is a natural preference for some viruses, such as reovirus, for tumor cells, whereas other viruses, such as Ads, VSVs, and HSVs, must be adapted to this selectivity for tumor cells. Healthy cells infected with OVs inhibit viral spread via different mechanisms that are frequently deficient in tumor cells [31]. The type I IFN-associated antiviral potential of a cell is correlated with the selectivity of a VSV for cancer cells. Normal cells produce type I IFNs to prevent viral replication; however, the majority of tumor cells have defective type I IFN production and are, therefore, proposed for VSV infection [40,68]

#### 3.2.4. Disrupting and Remodeling the Vascular System at the Tumor Site

The factors that influence tumor angiogenesis are represented by the expression of oncogene-mediated proteins and by factors that induce cellular conditions, such as low pH, hypoxia, nutrient deficiency, or induction of reactive oxygen species (ROS) [112]. The antiangiogenic modalities of OVs are as follows: (i) Direct infection of tumor cells followed by lysis of vascular endothelial cells of the newly formed vasculature; (ii) Immune response triggered by OVs, which leads to cell aggregation and slowing of blood flow; (iii) Expressing viral proteins that block the synthesis of factors promoting angiogenesis induced by the tumor (Figure 4) [10].

Many OVs are characterized by their direct antivascular properties. An adenovirus, OBP-301, containing elements of human telomerase reverse transcriptase that modulate the expression of E1 genes, was designed to produce IFN-gamma with antiangiogenic effects that cause selective destruction of cancer cells from murine colon tumors in syngeneic mice [113,114,115]. Within the blood vessels, VSVs stimulate clot formation and inflammation. VV infection results in vascular leakage and collapse by dysregulating the EGFR/Ras signaling pathways [97]. The OV iNDV3a-LP enhances endothelial cell lysis [116]. Armed VV-, HSV-, MV-, and Ad-based systems modulate endostatin and angiostatin levels and promote vascular collapse. In addition, VEGF can be downregulated by interacting with antiangiogenic proteins, followed by damage to new blood vessels in the TME, ultimately triggering oncolysis [117,118].

#### 3.2.5. Metabolic Reprogramming of Tumor Cells

Metabolic reprogramming in cancer cells promotes proliferation and delays antitumor immune responses. Cancer cells, unlike normal cells, undergo aerobic glycolysis or the “Warburg effect”, consisting of a high glucose uptake and glycolysis followed by lactic acid fermentation in the cytosol, the last product of glycolysis being pyruvate, which is converted to nicotinamide adenine dinucleotide (NAD)^+^. This alternative pathway is modulated by a blockade of pyruvate dehydrogenase (PDH) [119,120,121]. Malignant cells compensate for low levels of glucose in the tricarboxylic acid cycle by breaking down glutamine through a process called glutaminolysis, which increases the influx of acetyl-coenzyme A from the oxidation of fatty acids. This metabolic reconfiguration allows for the proper growth of neoplastic cells [122,123]. According to the “Warburg effect”, there is an increase in glucose intake and lactate secretion in malignant cells, which is characterized by accelerated metabolism by aerobic glycolysis, and which is 10–100 faster than oxidative phosphorylation, resulting in the synthesis of more ATP [124,125]. Cancer cells can use alternative metabolites, such as glutamine, for the synthesis of nucleotides, lipids, and proteins. This anabolic phenotype of malignant cells inhibits the function of many immune cells and contributes to the formation of immunologically “cold” cancers through a specific microenvironment containing low concentrations of nutrients and oxygen and increased levels of metabolic waste products. Most viruses hijack the metabolic pathways of the host cell to obtain resources for replication, such as amino acids, lipids, and nucleic acids [126]. Relationships between OVs and specific cancer metabolic pathways, including glycolysis, oxidative phosphorylation, cholesterol synthesis, pyruvate metabolism, glutaminolysis, and fatty acid oxidation, have been described. With regard to pyruvate metabolism, some OVs, such as VVs, influenza A (H1N1) viruses, hepatitis C viruses, and reoviruses, enhance the expression of inhibitory PDH kinases, followed by inhibition of PDH activity, thereby amplifying oncolysis [127,128]. ROS also play a role in viral replication and oncolysis. The vesicular stomatitis virus activates transcription factor NRF2 to promote oncolysis [129]. At the TME level, there is competition between tumor and immune cells for essential resources, such as glucose, tryptophan, and glutamine. Moreover, cancer cells produce immunosuppressive metabolites, such as kynurenine, lactic acid, and adenosine. In this metabolic context, an immunosuppressive microenvironment is promoted at the TME level, which is favorable for the polarization of macrophages toward the M2 phenotype, the development of Tregs, and reduction in CD8^+^ [130,131]. 

#### 3.2.6. Role of Immunogenic Cell Death (ICD) in Oncolytic Virotherapy

Some main types of immunogenic cell death, including autophagy, necroptosis, and pyroptosis, which are involved in the death of tumor cells infected with native oncolytic viruses such as Ads, VVs, NDVs, coxsackievirus B3, and MeVs, have been described [131].

It was demonstrated on HOS and A549 tumor cells that adenovirus serotype 5 initiates multiple cell death pathways, including inflammasome activity. Also, the Semliki Forest virus (SFV4) triggers rapid cell lysis accompanied by the induction of apoptosis. During tumor cell lysis, the vaccinia virus Western Reserve induces necroptosis and autophagy. Both AD- and SFV induce potent ICD in infected tumor cells, followed by the activation of DCs and antigen-specific T-cells. 

Dying cells release DAMPs in the TME, indicating the existence of a nonself entity and resulting in the induction of antitumor immunity [132,133]. DAMPs and neoantigen epitopes confer antigenicity to cancer cells, resulting in ICD [134]. A wide range of factors has been reported to be responsible for this canonical ICD. The most important factors are CRT, ATP, HMGB1, HSPs, and annexins [135]. One of the DAMPs that enhance the phagocytosis of antigens by DCs is CRT [136]. ATP is released from apoptotic cells and binds to the P2Y2 receptors on macrophages and monocytes, accelerating the elimination of apoptotic cells [137]. HMGB1 activates TLR4 in immature DCs, causing the maturation of DCs and accelerating the uptake of tumor antigens by DCs [138,139]. HSPs can bind to DC receptors stimulating the antigenic peptides; they also stimulate the intrinsic proinflammatory cascades [140,141,142]. Annexin A1 (ANXA1) plays an essential role in resolving inflammatory processes and assisting DC activity [143]. Other factors involved in the development of ICD include cytokines, chemokines, nucleic acids, and translational factors. Some OVs have a natural tropism for tumor tissues, whereas others are genetically modified to replicate in tumor cells to detect specific targets and deliver certain genes (Figure 5).

Immunogenic apoptosis is the most common type of OV-induced cell death. Receptors, such as TRAIL-R, TNF-R, and Fas, form a death-inducing signaling complex (DISC), are modulated by a viral infection, and regulate the activation of caspase-8, inducing apoptosis [144]. These receptors are hyperexpressed and trigger the caspase cascade, initiating extrinsic apoptosis [145]. Ads induce apoptosis in tumor cells, which is a natural feature. An H5CmTERT-Ad strain that expresses a trimeric tumor necrosis factor (H5CmTERT-Ad/TRAIL) exhibited a more powerful tumor-killing effect than natural Ads in a control arm [146]. The HSV is also involved in apoptosis. An engineered oHSV-2 was armed with an activator of apoptosis, Her2-COL-sFasL, to enhance caspase activation in infected cells and amplify apoptosis [147]. Necrosis represents premature cell death by autolysis, consisting of the loss of cell membrane integrity, swelling of organelles, and uncontrolled release of cell death products into the extracellular space [148]. Necroptosis is a process of programmed necrotic cell death that can induce or enhance antitumor immunity. It is frequently associated with cascading inflammasome activation as a result of the release of DNA, ATP, uric acid, nucleoproteins, and HSPs, which activate a kinase-like protein (MLKL), followed by membrane permeabilization and the release of DAMPs [149,150]. OVs also trigger multiple cell death processes in target neoplastic cells. For example, wild-type Ad-serotype 5 induces autophagy, pyroptosis, and necroptotic cell death pathways [131]. The NDV triggers apoptosis and was found to activate the intrinsic and extrinsic apoptotic pathways [12]. The oncolytic NDV strain FMW (NDV/FMW) is an ICD inducer in lung, melanoma, and prostate cancer cells, displaying HSP70/90 and ATP secretion and triggering apoptosis, autophagy, and necroptosis in melanoma cells [151,152,153]. NDV/Hitchner B1 promotes apoptosis and necroptosis in glioma cells [154].

## 4. Delivery Systems of OVs

### 4.1. Direct Delivery of Oncolytic Viruses

#### 4.1.1. Intratumoral Delivery

Intratumoral administration is the most frequently employed route for delivering OVs in clinical trials [37]. Various trials have established the effectiveness of intratumoral administration and studied its capacity to conserve adequate bioavailability of OVs by avoiding the circulatory torrent [155]. The local approach of intralesional administration benefits this process by decreasing viral inactivation. Limited viral interactions with the innate immune system reduce the likelihood of systemic toxicity and transfer of the desired viral load through a unique administration. The limitations include a nontherapeutic level achieved in the tumor owing to a dense ECM and an inadequate intratumoral infusion owing to neoangiogenesis [156]. Intratumorally administered OVs can induce a systemic response. The production of tumor antigens is a response to the interaction between OVs and tumor cells [157]. Studies on the administration of T-VEC have shown a decrease in the size of the studied melanoma lesions, both injected and noninjected [158].

#### 4.1.2. Intravenous Delivery

Intravenous administration has benefits for cancers in the metastatic stage when it is necessary to infect a varied number of tumors of different sizes and locations, but also in cases where intratumoral administration is not feasible owing to tumor location. This type of administration can ensure a better distribution of OVs in the tumor mass; however, the optimal OV dose is uncertain. Some OVs can stimulate a more pronounced immune response when administered intravenously [159]. The immune system can prevent the successful spread of OVs to target tissues using different strategies. OVs can connect nonspecifically to serum proteins or circulating cells from the circulating stream, with subsequent destruction [160]. As these mechanisms are part of the innate immune system, they are also effective in patients who are not exposed to a particular virus, and a history of exposure can cause significant destruction of the virus through the presence of a precise and vigorous immune response [161]. By binding to viral particles, antibodies can be eliminated through complement action or destruction by macrophages located in different organs. The liver, spleen, and lungs are the main organs involved in the neutralization of OVs [162]. Another barrier to the effective intravenous administration of OVs is the tumor ECM and interstitial fluid pressure. High interstitial fluid pressure limits access to the center of the tumor, where it has the highest value owing to the fluid flow at its periphery, but it also decreases the passage of large molecules [163].

### 4.2. Systemic Delivery by Cargo of Oncolytic Viruses 

There have been numerous and varied studies in the field of OV cargo delivery to tumors, which can be classified into biological and nonbiological delivery systems. Biological carriers include cellular, cell membrane-mediated, and serum albumin-mediated carriers of OVs [10]. Nonbiological carriers of OVs include mineralized nanostructures, magnetic nanostructures, and polymer-based nanocarriers. 

#### 4.2.1. Organic Carriers of OVs

Cellular Vectors

Mesenchymal stem cells (MSCs) have received increasing attention in scientific studies owing to their distinctive biological properties, wide therapeutic application, and impact on tissue engineering. MSCs exhibit this capacity by differentiating into osteogenic, adipogenic, or chondrogenic tissues and have a vast secretory network that includes multiple mediators, cytokines, and signaling molecules, in addition to their significant traits of self-renewal and multipotency [164]. This secretion shapes the inflammatory response and controls critical infiltration processes required for tissue regeneration and repair. The modulation behavior of MSCs is driven by feedback within the MSC molecule–target cell axis. MSCs are unable to display costimulatory particles and, consequently, are related to lower immunological activity, with the opportunity to be used in cell-based immunotherapy for tumors [165,166]. MSCs may facilitate the viral replication of oAds in vitro, enabling transportation and improving the long-term viability of viruses by preventing viral-specific immune responses in a rat model. Additionally, compared to virus inoculation alone in vivo, CR-MSCs may decrease the amount of IFN synthesized by stimulated T cells and leads to a greater increase in both the dispersion and survival of oAds [167]. Systemically administered MSCs infected with oAds in hepatocellular carcinoma xenograft specimens enhanced virion accumulation and significantly inhibited tumor proliferation [168]. Another study involving a replicative Ad carried by human MSCs, which included an E1A gene regulated by the alpha-fetoprotein promoter, showed significant growth restriction in orthotopic and subcutaneous hepatic xenograft cancer rodent models [169]. Following systemic delivery, MSCs contaminated with an oAd demonstrated therapeutic properties in orthotopic murine models of lung and breast malignancies, improving survival [164]. In experimental models of metastatic breast carcinoma, systemic injection of human MSCs engineered with copies of replicable viruses, such as Ad5/3.CXCR4 and Ad5R6D.CXCR4 Ads increased the survival of animals [170]. Systemic infusion of MSCs infected with a HER2-retargeted oncolytic HSV caused the spread of infection to breast and ovarian cancer cells in laboratory models, followed by a reduction in metastasis from the lungs and brain [171]. Research on SKOV3ip. One ovarian tumor xenograft has shown that intraperitoneally administered MSCs loaded with oncolytic MeVs in SKOV3ip. One ovarian tumor xenograft can pass into tumor lesions and ultimately transmit viral infection in mice, increasing the average lifespan [172]. CRAd5/F11 chimeric oAds targeting MSCs significantly inhibited the growth of colorectal cancer cells [173].

Neural stem cells (NSCs)

Some studies have shown that NSCs possess the ability to migrate into brain tumor cells and potentially provide a transport platform for various anticancer agents. An Ad (CRAd-Survivin (S)-pk7 virus) can target the CD46 receptors on NSCs. Infected NSCs migrate to glioma cells, thereby reducing tumor growth [174]. NSCs and MSCs were used as vectors to transport an oAd in a glioma model, and it was observed that both NSCs and MSCs permitted replication of the virus and decreased neuroinflammation. However, NSCs released a higher number of Ads than MSCs, and survival was prolonged using virus-loaded NSCs in tumor-bearing animals injected intratumorally, suggesting that the cause was hyperexpression of CXCR4 in the NSC cell line [175]. Kim et al. reported that the combination of CRAd-S-pk7 and NACA improved the replication of oAds in NSCs, thereby increasing the efficiency of the OV [176].

Cell Membrane-Mediated Systemic Delivery of OVs

Certain entities with potential as carriers are collectively available under the diverse umbrella of extracellular vehicles originating from several cell types. The benefits of employing these carriers, represented by biomimetic technology as an alternative to PEGylation, include safety and durability in the environment, thus overcoming the limitations of drug-delivery systems [177]. 

The notable advantage of integrating transporters derived from the cell membrane is largely due to the preservation of the inherent functionalities and signaling pathways within the cells from which they originate. Combined with the fact that these nanovehicles contain portions of the membrane that originate from only one source, such as the membranes of erythrocytes, platelets, cancer cells, and cells that are responsible for immune defense, or are generated from a couple of distinct cell types, the result is a valuable selection of carriers that possess the ability to target neoplastic cells, are the result of a bioengineering process of cell membranes, and can carry small compounds, such as genetic material, specific pharmaceuticals, and vaccine antigens. These vesicles generated from elementary components of the cell are used in oncological trials, mainly by transporting entities inside them or by enveloping nanoparticle–drug complexes [178]. Among all cell source alternatives, erythrocytes offer some of the most useful features for the objective in question. It is possible to obtain these cells fairly easily with a long average duration of survival in the vascular system, and there is also a potential to encapsulate an additional quantity of products because of the absence of natural cell organelles in these cells [177]. Novel models that rely on vesicles with a mixed coating, named erythroliposomes, generated from a combination of artificial components with the erythrocyte membrane, are beneficial for distribution [179]. Subsequently, neoplastic cell membranes were examined in vitro and in vivo to assess the extent of viral infection and its destructive impact. The particles, thus produced, known as ExtraCRAd, were delivered intratumorally and showed adequate absorption via clathrin-mediated endocytosis, which is an important benefit considering the demand for the expression of the CAR for infection in oAds. Furthermore, this setup presents an important limitation for antibody identification and inactivation in vivo [180].

Serum Albumin-Mediated Systemic Delivery of OVs

Wrapping OV particles in different materials is a strategy to protect them against the action of the immune system. The association between Ads and serum albumin prevents the action of antibodies against OVs. Antibodies recognize the capsid proteins of the virus and coat them via opsonization. Thus, they make OVs the targets of immune effector cells; however, they can also block transduction at the cellular level. Albumin is the protein with the highest concentration in the plasma and has a long half-life [181].

Following genetic engineering, albumin naturally binds to the surface of the Ad. The most common protein that forms the viral capsid has hypervariable regions that can facilitate the interaction between albumin and the virus but are also the main target of anti-OV antibodies. Increasing the viral load in the tumor bed can be achieved using proteolytic enzymes that degrade the ECM. Hyaluronan is one of the constitutive elements of the ECM. It is a complex polysaccharide present at high concentrations in some cancers. Hyaluronan accumulation can occur in the epithelium of digestive cancers, such as the colon and stomach [182,183]. Aggregation is evident in the tumor stroma of ovarian, breast, and prostate cancers. There are organs with rich hyaluronan pluristratified epithelium and stroma. In these cases, changes are visible and correlate with cancer progression. In squamous carcinomas of the head, neck, and skin, the hyaluronan levels are decreased and relate to the neoplasia stage [182]. Increased concentrations are associated with aggressive neoplasms and poor prognoses [184]. Hyaluronidase degrades the ECM and decreases interstitial fluid pressure. Consequently, the action of antitumor agents on target cells increases [185]. Hyaluronidase expression increases the effectiveness of OVs. Hexon is an essential protein found in the Ad envelope. The capsid protein binds to the binding domain of albumin. The proteolytic action of hyaluronidase and the mechanism of delivery based on the association of albumin with the surface of the virus determine the oncolytic potency of OVs [104]. VCN-11 is an adenovirus modified to express hyaluronidase and albumin-binding domains. The cytotoxicity and ability to evade the immune system are currently being investigated. These OVs have the most potent action on neoplastic cells and are 450 times more cytotoxic. In vivo, a high titer of neutralizing antibodies is present in the group in which OVs contain the albumin-binding domain, indicating a lack of their action [186]. VCN-01 is an oAd whose hyaluronidase acts on pancreatic desmoplastic stroma [187]. It can potentiate the effect of chemotherapeutic agents [188]. HAdV5, one of the best-characterized Ad types, targets the respiratory epithelial cells. Infection with this virus can be easily controlled in individuals with competent immune systems. This virus is a good candidate for oncolytic therapy, considering its individuality. This virus ensures efficient transduction into infected cells. Therefore, following the infection of the target cell, the genetic material is transferred into the nucleus. After the release of nucleic acids into the nucleus, cells produce the desired proteins using the genetic material provided by the virus. In addition, this method is used for dividing cell lines and cells that are more difficult to access, such as primary cells. Another advantage is the possibility of large-scale production [189]. Cells adapted to suspension culture ensure facile procedures for cell culture, thus increasing safety and lowering production costs [190]. By adapting the albumin layer to the virion, it is possible to reduce the removal of OVs in the liver. By attaching adapter proteins to this complex, the genetic material can be transmitted to neoplastic cells expressing HER2 or EGFRs. In addition, this adaptation reduces immune-mediated removal. Immune neutralization can occur through antibodies that block a specific receptor or expose it to proteasomal degradation; this is known as antibody-dependent intracellular neutralization. The innate immune system interacts with OVs through IgM antibodies and the complement system [189].

Extracellular Vesicles (EVs)

Extracellular vesicles are membrane vesicles released by various types of cells that are involved in intracellular signaling by carrying lipids, proteins, RNA, DNA, and carbohydrates and are classified according to their size as exosomes, microvesicles, apoptotic bodies, and oncosomes. They are frequently used for carrying anticancer drugs, such as OVs. Garofalo et al. reported that Ad5D24-CpG and PTX encapsulated in EVs induced peritumoral inflammation, increasing the efficacy and cytotoxicity in lung cancer models in vitro and in vivo [191,192]. Exosomes derived from tumor cells frequently have immunosuppressive functions; however, exosomes from tumor cells infected with OVs may have immunostimulatory functions. Labani-Motlagh et al. showed that melanoma cells infected with OVs, armed with CD40 (CD40L) and 4-1BB (4-1BBL) ligands, enhanced the activation of DCs and responses in the TME [193]. OVs can contain exosomes that selectively accumulate in tumors, expressing antitumor effects locally or in distant tumors [194]. They were designed to be LOAd oncolytic adenoviruses armed with immunostimulatory transgenes. LOAd-infected tumor cells expressing the transgenes packed into the endosomal compartments and released as exosomes. Also, mRNA was significantly increased in exosomes derived from both LOAd700- and LOAd703-infected cells. It was observed that any infected cell can express the transgenes even if the virus will only replicate in tumor cells. The immature DCs were then infected with the LOAd viruses and activated in vitro by the exosomes. OVs arm tumor-derived exosomes with their immunostimulatory transgenes, and this may partially explain why local treatment can result in systemic immunity. It was also demonstrated that the trafficking of microRNA (miRNA) into exosomes is altered in infected cells [193,195].

Microparticles (MPs)

Another vehicle used for the systemic transport of OVs is represented by microparticles (MPs), which are vesicles with a maximum diameter of 1 μm released from cells through exocytosis [196]. Microparticles confer immune protection and prolong the release of the oAds. Ads encapsulated in microparticles or PLGA conferred an immune protection for oAds [197]. High protection of oAds from inactivation using a system for total recirculation of MPs increases the release of oAds [198]. Tumor cell-derived MPs were used as cargo for oAds, which were delivered to the nucleus of tumor cells and protected Ads from antibodies; oAds carried by MPs represent a more effective treatment of tumors than treatment with free oAds [199].

#### 4.2.2. Inorganic Vehicle-Mediated Systemic Delivery of OVs

Various types of nanoparticles (NPs), including liposomes, dendrimers, gold nanoparticles, silica nanoparticles, and iron oxide nanoparticles, can be linked to OVs. There are two strategies for the delivery of OVs using NPs. The first is shielding and surface modification and the second is the combination therapy of OV-NPs. There are two mechanisms for delivering NPs to the tumor site, namely active and passive. In the passive mechanism, the EPR effect increases the permeability of tumor blood vessels, and NPs are concentrated in the tumor region [200]. Active delivery involves the selective binding of NPs to target cells with the aim of increasing the concentration of the therapeutic agent in neoplastic tumors with fewer side effects [201]. Various biological ligands, such as peptides, proteins, aptamers, and polysaccharides, exhibit affinity and specificity for molecules or receptors in tumor cells [202].

Shielding in Drug Delivery

Polyethylene glycol (PEG) is most frequently used to shield the surface of NPs owing to its biocompatibility, stability, and hydrophilicity in solutions with extreme pH [203]. PEG was successfully combined with Ad vectors, which extended the circulation time and reduced liver toxicity [204]. Arginyl-glycyl-aspartic acid (RGD) is another ligand that enters cells via endocytosis [205]. Cationic liposomes (DOTAP/DOPE) have been reported to enhance the transfer of viral genomes in tumor cells and increase the permeability and preferential targeting of tumors [206].

OV-NP Combination Therapy

AuNPs are widely used in oncolytic viral therapy, facilitating the access of DNA to the tumors despite the presence of antibodies [207]. A particle containing Ad coated with micelles and conjugated with paclitaxel was designed and demonstrated high efficiency against tumors [208]. We tested PEG-polylactic acid (PEG-PLA) in combination with polymer micelles to carry paclitaxel to specific targets, which induced a higher antitumor effect than paclitaxel [209]. Another polymer, polygalactosyl-b agmatyl diblock copolymer (Gal32-b-Agm29), which coats an oAd, demonstrated a very strong affinity for a glycoprotein receptor expressed on the surface of hepatic tumor cells. oAds can also be combined with hydrogel systems, which provide a protective environment that determines the possibility of cytotoxic effects on malignant cells [210]. oAds can be loaded in alginate gel and intratumorally administered and have demonstrated a long-acting efficacy of oAds compared with naked oAds in tumor xenograft models [211]. Almstätter et al. combined magnetic technology with OV delivery to improve the delivery of OVs. Four magnetic virus complex models have been designed, which were shown to improve the efficacy of OVs and reduce systemic toxicity [212].

## 5. Current Strategies for Oncolytic Virotherapy

### 5.1. Resistance to Oncolytic Virotherapy

Virotherapy does not elicit the same response in all patients and cancer types. The limitations and challenges of this treatment method, as well as the existence of therapeutic resistance, have been described [213,214]. Host immune response to viruses is a key challenge in virotherapy. In several preclinical studies, the host immune response has been demonstrated to play important roles in reducing replication and faster clearance of OVs and is responsible for decreased antitumor potency. In this context, new therapeutic strategies involving OVs focus on modulating the host immune system to maximize antiviral responses and consequently improve tumor destruction [9]. Different mechanisms of resistance have been described; these include antiviral immune, tumor cell-mediated, stromal cell, and systemic responses [214]. Resistance to viral infection can be accomplished in four stages: (i) Entry and binding of the virus; (ii) Viral transcription and translation; (iii) Cell survival; (iv) Cell signaling. Multiple factors, including the presence of neutralizing antibodies, normal cell receptors, and proteases in the ECM, influence the binding and entry of the virus. During epithelial–mesenchymal and mesenchymal–epithelial transitions, intercellular junctions are reported to be strengthened, which makes the penetration of Ovs difficult [215].

To perform the transcription and translation of its own genome, the next crucial step for the virus is to properly regulate host transcription and the machinery responsible for translating mRNAs into proteins. For example, tumor cells can influence viral transcription through epigenetic modifications and the expression of several proteins, such as the apolipoprotein B mRNA-editing enzyme. Several survival proteins protect tumor cells against viral infections. In addition to epigenetic modifications, resistance to OVs has been found to be associated with the modulation of autophagy, apoptosis in tumor cells, and promotion of cell survival. Li et al. demonstrated an increase in the oncolytic activity of virotherapy, where beclin-1 activity in Ads promoted tumor cell apoptosis. In addition, resistance to viral infection and multiplication in tumor cells has been associated with several cell survival regulators (MEK, MAPK, MYCN, MDR1, and SMAC) [108,216]. Infected cells release soluble signals that protect surrounding cells from various viral infections. Furthermore, triggering of the interferon-receptor (IFNR)-based Janus-kinase (JAK-STAT) pathway in tumor cells by the secretion of interferons in the context of infection with OVs has an important role in antiviral resistance [217]. The antiviral activity of stromal cells is stronger than that of tumor cells, which affects the therapeutic response to OVs. Arwert et al. showed that CAFs prevent viral infection through interferon regulatory transcription factor 3 (IRF3) and stimulator of interferon genes (STING). Similarly, in ovarian and breast cancers, tumor associated-macrophages promote resistance to OVs by secreting IFN, which creates a protective antiviral status [218]. OVs can also induce a local immune response, which subsequently converts an immunologically “cold” TME into a “hot” one, infiltrated by immune cells, where the interferon-signaling pathway plays an important role [214,215]. The mechanisms that interfere with the efficacy of virotherapy are present at different levels and involve interactions of different cells in the TME. Thus, epithelial cells represent a physical barrier that controls the spread of viruses in tumors. Moreover, tumor angiogenesis is a regulatory component of viral dissemination and infection. Another factor responsible for resistance to oncoviral treatment is the presence of hypoxic zones in the tumors. Hypoxia can decrease the proliferation of OVs in tumor cells and disturb the lytic mechanism of tumor cells. Hypoxia-inducing factor 1 (HIF-1α) is a main factor in hypoxia that promotes angiogenesis and metastasis of tumor cells [219]. In addition to physical constraints, prostaglandins, and VEGF are associated with resistance in the TME [214]. Difficulties in the development of novel therapeutics using viral vectors are associated with ineffective distribution, hypoxia in the tumor, vascular barriers, suboptimal immune responses, and antiviral molecular mechanisms that occur at the intercellular level. Taken together, certain mechanisms exert a degree of resistance to virotherapy and are responsible for the subsequent killing of tumor cells. Nevertheless, the role of this mechanism in conferring resistance to OVs remains unclear [214].

These data indicate the need to identify novel factors, such as tumor hypoxia and epigenetic modifications, which are considered to be the components responsible for resistance to oncolytic virotherapy. Additional data on this mechanism will provide researchers and clinicians with the perfect tools to create a rational design of OVs and devise appropriate therapeutic strategies [214].

### 5.2. Current Status of Combining Oncolytic Virotherapy with Other Synergistic Therapies in Cancer

As previously mentioned, several factors that contribute to the resistance to OVs during monotherapy have been described. Combination therapies have been considered to overcome this problem. This approach has demonstrated efficacy in various types of tumors in preclinical and clinical settings [35].

#### 5.2.1. Combination with Chemotherapy

Chemotherapy remains a conventional anticancer therapy. OVs act synergistically with chemotherapy at various levels, including the inhibition of the immune-mediated degradation and neutralization of the virus. Chemotherapy complements virotherapy through numerous mechanisms, such as through the enhancement of tumor cell immunogenicity, direct killing of cancer cells, and blockage of antiviral immune responses (NCT00006106, NCT00634595, NCT01584284). Cyclophosphamide is an alkylating agent that increases the efficacy of virotherapy using different viruses. The immunomodulation triggered by this chemotherapeutic agent is thought to improve the replication, spread, and cytolytic activity of the virus. Considering the promising results in preclinical and phase I and II studies (NCT00006106), an engineered Ad vector, ONYX-015, was administered in association with 5-fluorouracil and cisplatin to treat squamous cell carcinoma of the head and neck or esophagus in a phase III trial. The response rate was 79% in patients treated with combination therapy, compared with 40% in those treated with chemotherapy alone [215].

In addition, the antitumor effects were intensified when the Ad was administered in combination with different drugs, such as doxorubicin, gemcitabine, paclitaxel, irinotecan, and cyclophosphamide. Furthermore, in preclinical studies, a combination of temozolomide with OV was tested in patients with glioma and showed a good toxicity profile and improved survival [35,213]. 

However, the efficacy of OVs in combination with chemotherapy remains controversial and requires further investigation. Although some clinical trials have shown promising results, others involving large cohorts have reported the inefficacy of this combination regimen. For example, Eigl et al. demonstrated a median survival time of 19.1 months in patients diagnosed with metastatic castration-resistant prostate cancer who were treated with a combination of docetaxel and pelareorep, compared with a median survival time of 21.1 months in patients who received docetaxel (NCT01619813) [220].

#### 5.2.2. Combination of OVs with Radiotherapy 

Several studies have demonstrated that a combination of virotherapy and radiotherapy leads to promising therapeutic results. Although robust preclinical data exist, this effect was due to the synergistic activity between these two types of therapies despite the lack of a well-established mechanism. This combination resulted in improved anticancer activity. Radiotherapy-induced cellular modifications can increase the capacity of viruses to replicate and disseminate through tumors. Pokrovska et al. proved that when OV was combined with radiotherapy, tumor-specific cytotoxicity was induced through increased virus production [221]. In a phase I trial, a mutated HSV (G207), combined with radiotherapy for recurrent malignant glioma, showed the potential for a clinical response. In this study, six out of nine patients showed a partial response or stable disease, which proved to have a synergistic effect [35,213].

In another approach, virotherapy could increase tumor sensitivity to radiotherapy. A p53-expressing Ad induced greater sensitivity in infected p53-defective tumor cells to apoptosis induced by irradiation, followed by greater tumor inhibition, compared with the reduced apoptosis of tumor cells that did not benefit from radiotherapy [34].

#### 5.2.3. Combination of OVs with Immunotherapy

Despite the few reports in the literature on the combination of immunotherapy and virotherapy, the association of immune checkpoint inhibitors targeting PD-1 and/or cytotoxic T-lymphocyte antigen-4 (CTLA-4) with OV offers a new perspective in cancer treatment, particularly in cancers with an immunosuppressive microenvironment in which monotherapy with PD-1/PD-L1 inhibitors can become ineffective. The oncolytic activity of this therapy determines a release of cytokines and other molecules, which have the effect of converting a “cold” tumor to a “hot” one, a modification that could improve the effect of immunotherapy on the susceptible target [213]. In addition to PD-1, a number of immune checkpoint molecules that are also expressed on TILS have been shown to induce an immunosuppressive microenvironment and to reduce activated CD8^+^ cells. This category includes CTLA-4, TIGIT, TIM-3, and LAG-3. OVH-aMPD-1 acts synergistically with anti-TIGIT and substantially improves immune responses in MC38 and Hepa 1-6 in tumor models [222,223].

Nevertheless, in refractory lung cancer, vvDD monotherapy in combination with anti-PD-1 and anti-TIM-3 mAbs failed to show a therapeutic advantage. Importantly, combining more than two ICIs as a therapeutic strategy can yield better antitumor results [10].

T-VEC is an intralesional therapy that is a leading combination of immunotherapies. The role of the GM-CSF in this association is to enhance cellular immunity. The action of T-VEC results from a combination of a direct oncolytic effect of the viral vector and an immune response trigger. Promising results of objective and enduring response rates were observed in a phase I trial combining T-VEC with anti-CTLA4, ipilimumab, for the treatment of stage IIIB-IV melanoma. Furthermore, in an open-label phase II trial in both primary and metastatic tumors, the objective response rate substantially increased with T-VEC in combination with ipilimumab as opposed to that in ipilimumab monotherapy. In a phase II trial of T-VEC with ipilimumab vs. ipilimumab alone for advanced melanoma, in a 5-year analysis, the median PFS was 13.5 months with the combination and 6.4 months with ipilimumab, and estimated 5-year OS was 54.7% in the combination arm compared with 48.4% in the ipilimumab arm (NCT01740297) [224].

Pexastimogene Devacirepvec (Pexa-VEC) is an oncolytic vaccinia virus that has demonstrated encouraging results in a phase Ib trial in combination with an anti-PD-1 inhibitor, cepilimumab, in patients with renal neoplasms. An overall response of 37.5% was reported for this combination therapy [9,213].

Adenovirus DNX-2401 is an OV that was investigated in a dose-escalated study, which included patients diagnosed with malignant glioma [225]. In that study, a single dose of DNX-2401, administered intratumorally, obtained variable tumor responses. Currently, this OV is being investigated in a phase II trial in combination with pembrolizumab in patients with glioblastoma. An interim analysis revealed a median overall survival of 12 months and a response rate of 47% in 48 patients treated with the combination therapy [213].

The idea of triple therapy combining anti-PD1/PD-L1 with anti-CTLA4 and an OV has recently become very attractive, with a study in which an Ad in combination with anti-PD-L1 and anti-CTLA4 significantly inhibited tumor growth and prolonged survival in a TNBC orthotopic mouse model. A synergistic increase in the antitumor effect through the enhancement of the proportion of CD8^+^ T and T memory cells and the depletion of Tregs and tumor-associated macrophages polarized to the M1 phenotype, which modulated the TME, was reported [226]. This combination therapy has great potential and represents a promising strategy for improving cancer cell therapy that should be available shortly. Considering the safety, toxicity profile, and promising results, a growing number of ICIs and OVs are entering clinical trials.

#### 5.2.4. Role of Kinase Inhibitors

Pharmacoviral approaches are among the strategies for overcoming resistance to OVs, which associate virotherapy with molecular therapeutics that target kinases. In preclinical models, important synergistic results have been obtained, as is evident from a major augmentation in the replication and dissemination of OVs in tumor tissues [227]. Receptor tyrosine kinases (RTKs), which are localized on the cell surface, control intracellular responses that mediate cell growth and survival in most cases. Some pathways, such as the MAPK cascade or the PI3K/Akt/mTOR pathway, are activated downstream by receptor-specific ligand binding. Hyperactivation of RTKs leads to uncontrolled cell growth and is an important drug target. Recently, a combination of virotherapy with RTK inhibitors has achieved some attractive effects [9,227]. High tumor interstitial fluid pressure (IFP) represents an obstacle to efficient drug delivery and is thought to limit the clinical benefits of OVs by impeding the access and intrusion of viruses throughout the tumor microenvironment. The reasons for the increase in IFP in tumor tissues are debatable. One mechanism that is supposed to contribute to this is the aberrant VEGF-mediated enhancement of vascularization. Kottke et al. noted that when combining a VEGF ligand, VEGF165, with oReoV, the administration of VEGF165 before OV could deform the tumor vasculature and improve OV delivery to the tumor tissues. Two tyrosine-kinase inhibitors, erlotinib and axitinib, were combined with oHSV1 to stimulate viral delivery by inhibiting oHSV1-angiogenesis and were reported to affect tumor growth and angiogenesis in pancreatic tumors in mice [227,228]. Collectively, these observations confirmed the rationale for combination therapy with different kinase inhibitors, leading to promising results in preclinical studies. Furthermore, the outcomes of preclinical studies are consistent with the idea that a pharmacoviral combination of OVs with these molecules can stimulate the oncolytic efficacy of virotherapy in cancer cells [227].

#### 5.2.5. Combination with Cell Therapy

Chimeric-antigen receptor (CAR)-T therapy represents a novel and promising treatment for cancer that produces effective and durable clinical responses. Synthetic receptors that redirect lymphocytes, principally T cells, eliminate cells expressing a specific target antigen. In 2017, anti-CD19 CAR-T cell therapy was approved by the FDA for the treatment of B-cell malignancies. There are some limitations of CAR-T cell therapy, including major toxicities, which limit its efficacy against solid tumors, antigen escape, limited persistence, and poor tumor infiltration. These have led to new approaches in which CAR-T therapy has been associated with other anticancer treatments to improve antitumor efficacy and diminish toxicities [229].

Anti-CD19 CAR-T cell therapy has revolutionized the treatment of B-cell malignancies. Several studies have demonstrated that CD-19 is expressed on the surface of multiple OV-infected tumor cell types, harboring the CD19-T coding gene (OV19T). Consequently, CD-19 stimulates CAR-T cells and triggers tumor lysis, stimulating the release of OV19T from dead tumor cells that can activate endogenous antitumor immunity [229]. Moreover, virotherapy can be concurrently combined with immunotherapy and CAR-T cells to increase the number of immune cells in the TME and improve the antitumor response. For example, Ads expressing IL12 can be administered simultaneously with systemic HER-2- and PD-L1-blocking antibodies. In preclinical tumor models, this association exhibited enhanced or synergistic effects. Moreover, the timing of the administration of anti-CTLA4, virotherapy, anti-PD-(L)1 antibody, and emerging ICI antibodies represents another important area that requires further observation. To increase the efficacy and improve tumor responsiveness, systemic exploration of the effects of immunotherapy administered before, after, and concurrently with OVs and CAR-T cells is required [213,229].

## 6. Conclusions and Future Prospects

The advantages of oncoviruses in destroying tumor cells stem from their qualities per se, which include direct cytolysis, the targeting of the immune system, the modulation of the TME, and direct features, such as modulation of specific pathways in the tumor cell, antiangiogenic action, and modulation of tumor cell metabolism. Genetic engineering has offered huge advantages in terms of the potentiation of cytolytic action and the diversity of tumor cell attack. The means of transporting the virus to tumor cells remains a challenge, with multiple possibilities for obtaining OVs exhibiting more efficient action. In the last few years, numerous studies have demonstrated promising results using nonbiological or biological nanovectors to transport OVs into tumor cells. The use of OVs as a single-agent therapy or in combination with other therapeutic strategies has been demonstrated in both preclinical and clinical studies. OVs have an antitumor potential; however, some specific obstacles remain in their development as a new class of anticancer treatment. The barriers that induce the failure of oncolytic virotherapy are inadequate OV penetration and spread, host antiviral immunity, individual differences among patients, the interactions of the oncovirus with virus-resistant or virus-sensitive tumor cells, and low or moderate efficacy when used as single agents. The efficacy of a single OV administration is low because the host immune system may clear it from the blood flow, the virus may express the transgene only for a short period, or viral spread may be limited. Studies have been conducted in mice treated with oHSV or oAds, in which viral particles are cleared quickly [20,230]. Moreover, oAds demonstrated limited efficacy and low persistence in target tissues. To overcome these limitations, oncolytic virotherapy can be combined with other anticancer treatments, such as chemotherapy, radiotherapy, targeted therapy, or immunotherapy. Therapeutic outcomes are related to the balance between the antiviral response of the host and OV-induced antitumor immunity; however, the biological mechanisms affecting this balance remain unclear. Various tumor types and stages of cancer remain challenging with regard to the selection of OVs, transport modalities, and their association with various systemic treatments. 

## Figures and Tables

**Figure 1 ijms-25-01180-f001:**
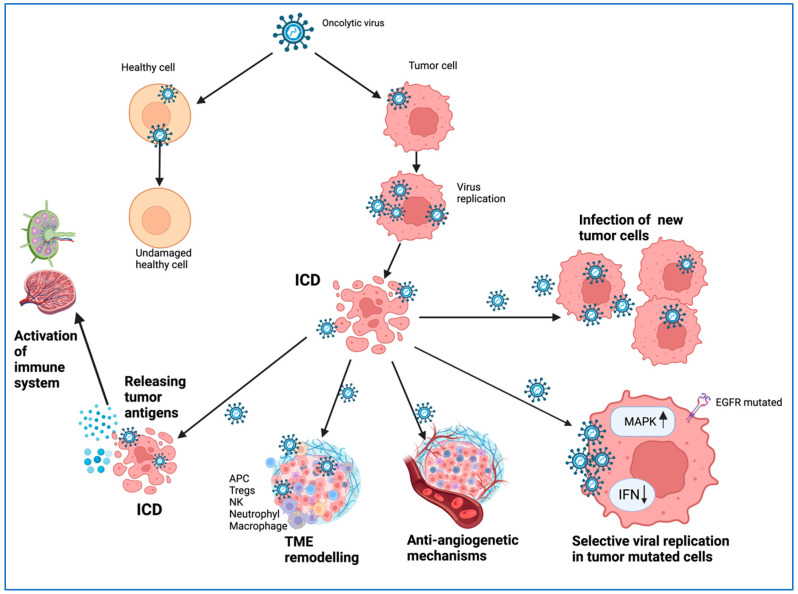
Mechanisms of action of oncolytic viruses. The replication of virus inside the tumor cell triggers immunogenic cell death (ICD), followed by release of tumor antigens that activate the immune system, remodeling of TME, activation of the antiangiogenic mechanisms, increase in viral replication in tumor-mutated cells, and spread of the viral infection to other tumor cells [9,55,56,57,58] (BioRender.com).

**Figure 2 ijms-25-01180-f002:**
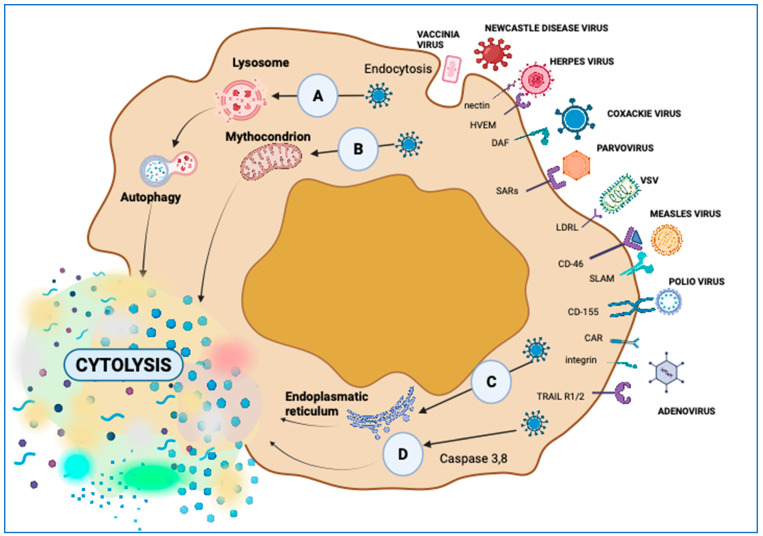
Oncolytic viruses (OVs) enter the tumor cell through endocytosis or by targeting one or multiple receptors. The mechanisms of direct oncolysis by OVs are the following: (**A**) Copious OV replication, followed by damage of lysosomes, autophagy, and cytolysis; (**B**) OV alters the permeability and mitochondrial metabolism, which is followed by cytolysis; (**C**) OV determines the tumor cell stress response, triggering the JNK pathway and caspase cascades, which is followed by cytolysis; (**D**) Protein on OV surface binds to TRAILR1 and TRAILR2 receptors, activating the TRAIL-mediated apoptosis (BioRender.com).

**Figure 3 ijms-25-01180-f003:**
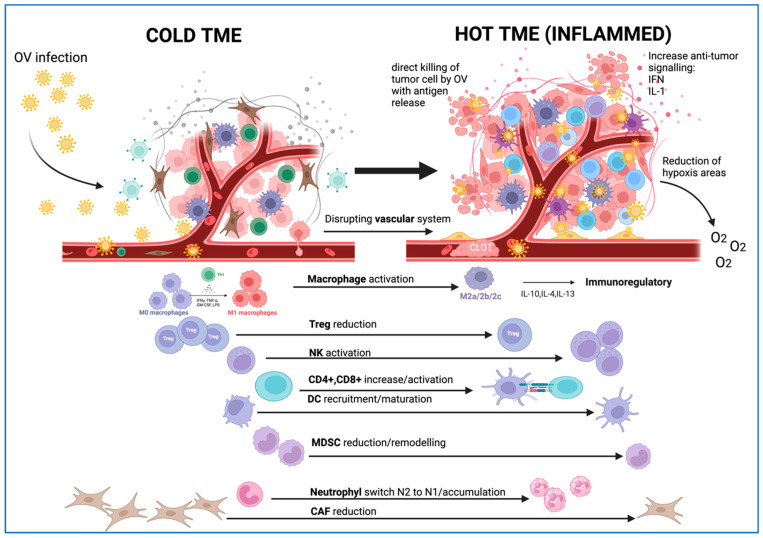
The cold TME contains stroma and immunosuppressive cells. OVs cause cell death. Remodeling of suppressive immune and metabolic environment to hot TME. CAF, cancer-associated fibroblast; DC, dendritic cell; IFN, interferon; IL, interleukin; MDSC, myeloid-derived suppressor cells; NK, natural killer; OV, oncolytic viruses; TME, tumor microenvironment; Treg, regulatory T cells (Biorender.com).

**Figure 4 ijms-25-01180-f004:**
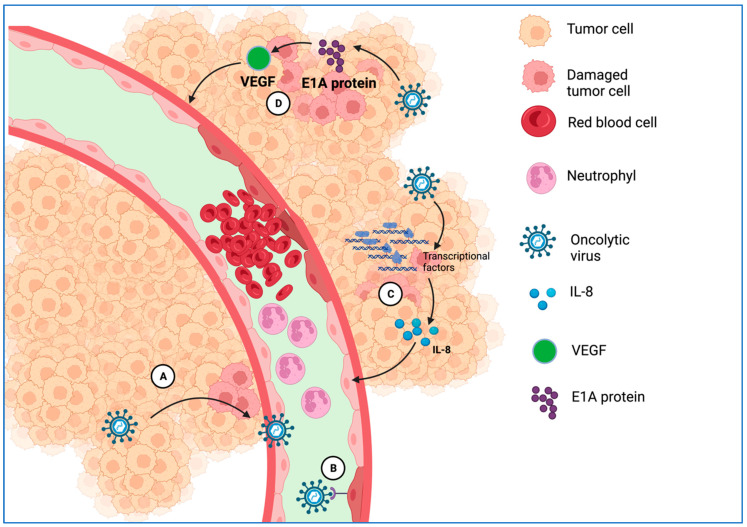
Antiangiogenic mechanisms induced by oncolytic viruses (OVs). (**A**) OVs determine the recruitment of neutrophils and the development of microthrombosis and ischemia, followed by apoptosis. (**B**) Engineered virus can induce direct lysis of endothelial cells. (**C**) OVs can modulate transcriptional factors, decrease the production of IL8, and, consequently, stimulate apoptosis. (**D**) OVs interact with angiogenic proteins and downregulate the expression of VEGF.(BioRender.com).

**Figure 5 ijms-25-01180-f005:**
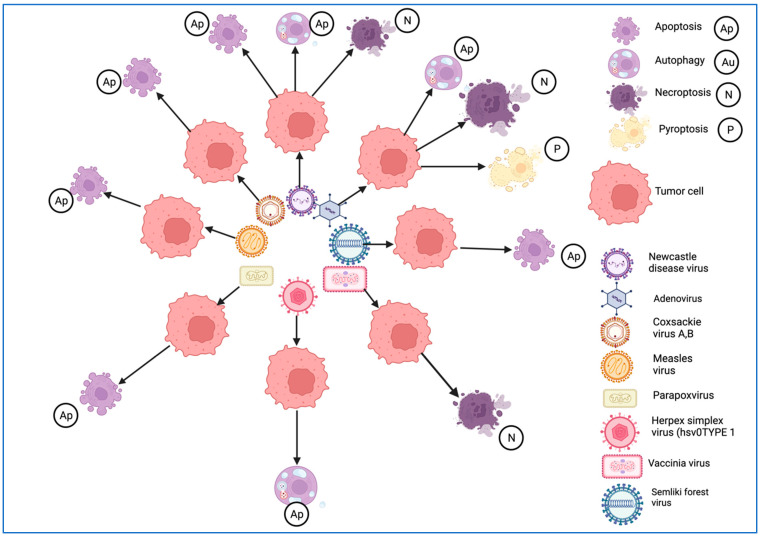
Immunogenic cell death pathways induced by natural oncolytic viruses (OVs). Some natural OVs, including Newcastle disease viruses (NDVs), coxsackieviruses (CVs), measles viruses (MVs), herpes simplex viruses (HSVs), vaccinia viruses (VVs), adenoviruses (Ads), semliki forest viruses (SFVs), and parapoxviruses (ORFVs) can induce a specific mode of immunogenic cell death. NDV, HSV, and Ad can induce multimodal cell death, modulating the exposure of damage-associated molecular patterns in dying malignant cells, followed by recruitment of innate immune responders and triggering of a strong immune response. VV induces necroptosis, and CV, MV, SFV, and ORFV induce immunogenic apoptosis (BioRender.com).

## Data Availability

Not applicable.

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
