# Peer review of "Oncolytic Virotherapy: A New Paradigm in Cancer Immunotherapy"

_ijms, 2024, doi:10.3390/ijms25021180_

Round 1
Reviewer 1 Report
Comments and Suggestions for Authors
I would like to congrate the Authors on their huge work.
Here are some suggestions.
1. Table 1.
May you structure Table 1, by putting the viruses names in column?
Since you reported a lot of information in Table 1, you may consider dividing it in two tables, one containing virus characteristics, one containing virus pathogenicity.
May you report in the table the same viruses (with the same names) you described in the text? (The same for lines 183-188.)
Lines 227-231. May you add the citation?
Figure 2. It lacks the letter "D".
Author Response
Dear Reviewer, Thank you very much for taking the time to review this manuscript. Please find the detailed responses in the manuscript re-submitted attached

Reviewer 2 Report
Comments and Suggestions for Authors
The manuscript by Volovat et al. is a very interesting review describing the role of viruses in the treatment of malignancies, especially cancer. The article is well written and organized, figures are appealing and the great number of data reported in the text makes this manuscript a valuable contribution in the field. Only few corrections in style and contents should be addressed. Hence, the review is suitable for publication after the following minor revisions:
- Authors refer to Figure 2D both in the text and in the caption, however in the figure a section D is not reported/clearly identifiable.
- Check the numbering of paragraphs in the whole manuscript. In line 232, the paraghraph should be 3.2.1; in line 255, should be 3.2.2 and so on.
- At the end of paragraph 4.2.1.4, authors mention the “antitumor effects locally or in distant tumors” induced by extracellular vesicles without specifying more details. Please give more insights about these aspects, describing the different pharmacological impliation of this type of treatment.
- Lines 796, 818 and 862, when discussing clinical trials, please add the NCT number to facilitate the consultation of the study from ClinicalTrials.gov.
- At the end of line 94, when introducing HSV, no literature references are reported. Please add some recent article in the field such as: Int J Mol Sci. 2023 Apr 11; 24(8):7092. doi: 10.3390/ijms24087092; Cureus. 2023 Mar 9;15(3):e35958. doi: 10.7759/cureus.35958. Similarly, appropriate citations should be added at the end of line 140, 537, 613 and 620.
- Rephrase lines 585-591
Author Response
Dear Reviewer, Thank you very much for taking the time to review this manuscript. Please find detailed responses in the manuscript re-submitted attached.

Reviewer 3 Report
Comments and Suggestions for Authors
I recommend Major revision:
Abstract:
- Consider specifying which types of cancers the OVs have been successful against in clinical trials.
- Expand on the types of standard treatment modalities that could be combined with OVs for better results.
- Change "there are several elements of resistance to the per se action of viruses" to a clearer phrase like "various factors contribute to resistance against the direct action of viruses."
- Consider rephrasing the sentence "This review provides comprehensive update on OVs" to "This review provides a comprehensive update on OVs."
- Define "per se action" for clarity.
- Add specific examples of standard treatment modalities being combined with OVs for increased response rates.
Section 1: Introduction:
- Provide specific examples or references for the case reports mentioned in the introduction.
- Specify the criteria for a virus to be considered "proimmunogenic."
- Clarify the sentence "it must be proimmunogenic, should not cause a chronic infectious disease" by specifying the characteristics that contribute to a virus being "proimmunogenic."
Section 2: Attractive Viruses for Oncolytic Virotherapy:
- Elaborate on the term "tumor tropism" for better understanding.
- Include information on the progress of clinical trials using the mentioned OVs.
- Change "Knowledge of the tumor tropism of different types of viruses" to "Understanding the tumor tropism of different viruses."
Section 3: Anticancer Mechanisms of Oncolytic Virotherapy:
- Specify which viruses fall under the category of RNA and DNA viruses.
- Add a brief explanation of the term "genetic stability" in the context of OVs.
- Elaborate on the mechanisms of viral replication and expansion for better clarity.
- In the sentence "Two-third of viruses in clinical trials have been reported," change "two-third" to "two-thirds."
- Consider adding a brief explanation of how different viruses target receptors on cancer cells.
Section 3.1: Selection of an OV:
- Clarify what is meant by "low immunogenicity with attenuated antiviral responses" and provide examples.
- Elaborate on why DNA viruses are more suitable for genetic engineering compared to RNA viruses.
- Change "relationshps" to "relationships."
Section 3.2: Antitumoral Mechanisms of OVs:
- Add more details on how OVs cause intracellular aberrations within pathways and gene expression in cancer cells.
- Provide specific examples or references for the mechanisms mentioned in Figure 1.
- In the sentence "considered immunologically 'cold' environments," consider rephrasing for clarity, e.g., "environments considered immunologically 'cold.'"
- Provide references for the studies mentioned, especially those demonstrating the immunomodulatory role of engineered OVs.
DC/T-cell Activation
- The section is generally clear but could benefit from a more structured flow.
- Consider breaking down the information into subheadings for better organization, such as "OV-Induced DC Activation," "Release of DAMPs," and "Genetic Engineering of OVs."
- Specify the acronyms the first time they are used (e.g., ICD, PAMPs, DAMPs, TME).
- Include references for specific claims, especially when discussing preclinical data (e.g., GM-CSF-armed HSV [56]).
- Ensure consistent use of terminology (e.g., "OVs" vs. "oncolytic viruses").
- Consider simplifying complex sentences for better readability.
Involvement of Macrophages
- Clarify the abbreviation "RV" – it is used without prior definition.
- Provide references for studies mentioned (e.g., studies on OVs, including VSV, HSV, RV, paramyxoviruses).
- Check sentence structure for clarity, especially in the latter part of the paragraph.
Involvement of NK Cells
- Specify the type of cancer models used in the studies for better context.
- Reference the study that demonstrated the survival prolongation with VVΔTKΔN1L.
- Consider breaking down complex sentences for improved readability.
Involvement of Neutrophils
- Provide references for key findings (e.g., the study on VV- and VSV-based viruses promoting apoptosis in tumor cells [68]).
- Simplify and break down complex sentences for better comprehension.
Involvement of Tregs
- Reference the study reporting oncolytic VV directly infecting and decreasing Treg numbers in murine cancer models.
- Ensure clarity in conveying complex information.
Stromal Modifications
- Provide references for studies on oAd expressing relaxin, PH20 hyaluronidase, ICOVIR15, and WRDD targeting CAFs.
- Maintain clarity in conveying scientific information.
Gene Targeting by OVs
- Clarify the statement about the natural preference of some viruses (e.g., reovirus) for tumor cells.
- Improve sentence structure for better flow and comprehension.
Disrupting and Remodeling the Vascular System at the Tumor Site
- Reference the study on OBP-301 containing elements of human telomerase reverse transcriptase.
- Ensure clarity and consistency in terminology.
Metabolic Reprogramming of Tumor Cells
- Provide references for the relationships between OVs and specific cancer metabolic pathways.
- Simplify complex sentences for better understanding.
Role of Immunogenic Cell Death (ICD) in Oncolytic Virotherapy
- Specify the types of oncolytic viruses (Ad, VV, NDV, coxsackievirus B3, and MeV) inducing immunogenic cell death.
- Break down complex sentences for better readability.
Moderate editing of English language required
Author Response
Dear Reviewer, Thank you very much for taking the time to review this manuscript. Please find detailed responses in the manuscript re-submitted attached

Round 2
Reviewer 3 Report
Comments and Suggestions for Authors
Acceptable for publication.